# The Role of Magnetic Nanoparticles in Cancer Nanotheranostics

**DOI:** 10.3390/ma13020266

**Published:** 2020-01-07

**Authors:** Maria Ferreira, João Sousa, Alberto Pais, Carla Vitorino

**Affiliations:** 1Faculty of Pharmacy, University of Coimbra, 3000-548 Coimbra, Portugal; ines.ferreira6556@gmail.com (M.F.); jjsousa@ff.uc.pt (J.S.); 2Coimbra Chemistry Center, Department of Chemistry, University of Coimbra, 3004-535 Coimbra, Portugal; pais@qui.uc.pt; 3Centre for Neurosciences and Cell Biology (CNC), Faculty of Medicine, University of Coimbra, 3004-504 Coimbra, Portugal

**Keywords:** cancer, therapy, diagnosis, magnetic nanoparticles, nanotheranostics, biomedical applications, drug delivery

## Abstract

Technological development is in constant progress in the oncological field. The search for new concepts and strategies for improving cancer diagnosis, treatment and outcomes constitutes a necessary and continuous process, aiming at more specificity, efficiency, safety and better quality of life of the patients throughout the treatment. Nanotechnology embraces these purposes, offering a wide armamentarium of nanosized systems with the potential to incorporate both diagnosis and therapeutic features, towards real-time monitoring of cancer treatment. Within the nanotechnology field, magnetic nanosystems stand out as complex and promising nanoparticles with magnetic properties, that enable the use of these constructs for magnetic resonance imaging and thermal therapy purposes. Additionally, magnetic nanoparticles can be tailored for increased specificity and reduced toxicity, and functionalized with contrast, targeting and therapeutic agents, revealing great potential as multifunctional nanoplatforms for application in cancer theranostics. This review aims at providing a comprehensive description of the current designs, characterization techniques, synthesis methods, and the role of magnetic nanoparticles as promising nanotheranostic agents. A critical appraisal of the impact, potentialities and challenges associated with each technology is also presented.

## 1. Introduction

Cancer diagnosis and treatment have improved over the last years, leading to increased survival rates. Conventional strategies including surgery, radiotherapy, chemotherapy and additional methods such as immunotherapy and hormone therapy have found success in cancer treatment. However, limitations such as post-surgery relapses, therapy inherent adverse effects, benefit restricted to a limited group of patients and low response rates have raised great concerns [1,2,3].

Despite the technological research and development, cancer remains a major global impact disease [4]. Current data from Globocan states that the 2018 incidence reached 4,229,662 new cases in the same geographic area, reporting 1,943,478 deaths, for all cancers. Europe comprises 23.4% of the worldwide cancer incidence and 20.3% of the registered mortality [5]. As a conjunction of multifactorial neoplastic diseases associated with elevated mortality rates, cancer presents extensive relevance in the medical, social and economic fields, which is accompanied with higher patient expectations and more strict requirements [1,5].

In response to these requirements, nanotheranostics has emerged as a promising field of study, using nanotechnology in the development of integrated diagnostic and therapeutic systems for biomedical application.

More specifically, magnetic nanoparticles constitute recent multifunctional platforms (with both imaging and therapeutic functions) in the nanotheranostic field [6,7]. The physical properties of these nanosystems enable their usage as imaging probes for cancer diagnosis and as drug delivery systems, which represent an appealing approach to monitor the real-time therapeutic response, useful for targeted therapeutic regimes and personalized medicine.

## 2. Magnetic Nanoparticles

Magnetic nanoparticles (MNPs) are a group of nanoparticles that exhibit magnetic susceptibility and thus, can be oriented and controlled by an externally applied magnetic field. MNPs can be composed of diverse materials, based on the exhibited magnetic effect, and depending on their orbital and spin characteristics. This is reflected in their magnetic susceptibility, which is determined by the variation of the nanoparticle’s (NP) magnetization in the function of the externally applied magnetic field [8,9]. Under these conditions, the various NPs will display rather different behaviours. Materials can present diamagnetic, paramagnetic, ferromagnetic, antiferromagnetic, ferrimagnetic or superparamagnetic properties, whether they respond to a magnetic field by developing negative or positive magnetization, presence or absence of remnant magnetization once the magnetic field is removed, temperature-dependent magnetic properties and spontaneous magnetization. Generally, a diamagnetic material responds to a magnetic field by developing a negative magnetization, in which there is a partial alignment of the magnetic moments in the opposite direction of the externally applied field. Under the same circumstances, a paramagnetic material develops a positive magnetization due to unpaired electrons in partially filled orbitals and partial alignment of the magnetic moments in the field direction. However, once the magnetic field is removed, both diamagnetic and paramagnetic materials have no interaction between the atomic moments and consequently, the magnetization is null. In contrast, ferromagnetic materials present strong interactions between atomic moments, with a parallel spin alignment in the presence or absence of an external magnetic field, when below the Currie’s temperature (T_C_). Thus, ferromagnetic properties are also temperature-dependent and enable spontaneous magnetization. On the same conditions, antiferromagnetic materials obey to an antiparallel spin alignment, whilst under Neel’s temperature (T_N_). Both exhibit an isotropic spin-exchange interaction, with no predetermined ideal orientation. Similar to anti-ferromagnetism and ferromagnetism, ferrimagnetism exhibits an antiparallel spin alignment, under T_C_. However, the magnetic ordering occurs in crystal structures, e.g., in ionic compounds, therefore the spin moments do not cancel each other, and the spin orientation follows a crystallographic axis, according to the crystal anisotropy [10,11]. The stability of the magnetization within the material also relies on the size and orientation of the magnetic domains, in which the spin collinearity and rotation are uniform. Thus, a multidomain ferromagnetic particle with nonuniform magnetization can be reduced to a size in which it becomes an oriented single-magnetic-domain particle (SDM) with elevated resistance to a demagnetizing field and opposing spin rotation. This phenomenon is represented by an elevation of coercivity. Under a critical diameter, the SDM suffers spin reversal due to thermal fluctuations that culminate in a particle of zero coercivity and no retained memory of the applied field, once this is removed. These particles exhibit zero coercivity and zero hysteresis, thus possessing superparamagnetic properties [10,12].

In the superparamagnetic state, MNPs demonstrate high magnetic susceptibility and rapid response to an externally applied field. Thus, for biomedical applications, including magnetic resonance imaging (MRI), magnetic hyperthermia, magnetofection, tissue repair, immunoassay, detoxification of biological fluids and cell separation, NPs integrating superparamagnetic materials at room temperature are considered the most effective [12,13,14,15].

In what pertains to magnetic materials, MNPs have been mentioned in the literature as composed of iron, cobalt, nickel, and presented in mixtures containing, for example, cobalt-platinum NPs, iron-platinum NPs, manganese, nickel or cobalt-ferrite NPs, magnetite (Fe_3_O_4_) or maghemite (γ-Fe_2_O_3_) NPs. The magnetic effect varies as a function of size, type and composition of the particles, as shown in Figure 1 and Table 1 [16].

Additionally, MNPs may be presented in various constructs: in the form of single-phase particles, core-shell NPs of two phases and polymer coating, multicore NPs or oriented chain arrays [16].

Among the diverse magnetic constructs available for multifunctional purposes in cancer theranostics, MNPs are predominantly structured with a magnetic core, shell and a polymer coating that upholds the targeting and therapeutic moieties [7,17]. The rationale behind this selection is based on the potential of the magnetic core, due to its intrinsic properties, to perform as agent for MRI contrast, hyperthermia therapy and/or controlled drug delivery and based on the functionality of the shell components to protect the core and overcome biocompatibility and immunogenicity limitations. The incorporation of a polymer coating prevents NPs aggregation, and consequently, provides an extended half-life, assists in controlled drug release and, simultaneously, provides functional groups for biomolecule conjugation. On this basis, functionalization agents can be added, such as chemotherapeutic (single or multiple drug therapy) and biotherapeutic agents (such as nucleic acids or proteins), targeting agents, fluorescent agents and/or photosensitizing agents, the latter with application in photothermal (PTT) or photodynamic therapy (PDT) and other functional features for cellular trafficking [18].

### 2.1. Synthesis

An extensive variety of synthesis methods have been implemented for the development of MNPs, using physical or chemical synthesis approaches. Accordingly, methods such as mechanical attrition, thermal decomposition, hydrothermal synthesis, sol-gel reaction, microemulsion, co-precipitation, sonolysis, electrochemical method and polyol method are some of the most relevant mentioned in the literature [17,19,20].

The selected process and protocol for MNPs synthesis determine their properties, and consequently, their applicability and performance. 

The physical synthesis strategy recurs mainly to the mechanical attrition method, thermal quenching or pyrolysis. For the implementation of the attrition method, mechanical force is set forth using planetary ball or media mill in order to transform the starting materials into nanosized particles. The thermal quenching is a strategy which combines a rapid quench process to produce amorphous components and subsequent thermal treatment for controlled size crystallization. In turn, the pyrolysis approach uses a high-pressure organic gas/liquid precursor forced through a cavity, that when burned into ash, originates oxidized magnetic nanoparticles [21]. These constitute widely established approaches that are considered viable options in what concerns mass production. However, some disadvantages regarding the homogeneity of size distribution may arise and compromise the MNPs behaviour.

Chemical synthesis is a more recent approach that involves a panoply of methods for MNP production, providing more favourable processes in terms of particle dimension, size distribution range, crystallinity and stability.

The classical approach is described by the LaMer and Dinegar model [22] of burst nucleation and growth. It consists of a three-step process that first involves the increasing concentration of monomers in a liquid media, and the burst reaction for the formation of nuclei, followed by the attachment of the monomers to the formed nuclei, originating monodisperse NPs [22,23,24].

The particles are initially involved in a reduction reaction and collision of ions and atoms in the presence of reducing agents, such as sodium citrate, sodium borohydride, molecular hydrogen, hydrazine, formamide, formaldehyde, polyaniline or ascorbic acid, among others, allowing the formation of nuclei. The selected reductant has an important impact in the final particle, as the stronger the reductant, the faster the nucleation reaction. On the other hand, weaker reductants are responsible for slower reaction conditions, thus enabling a more controllable size and shape of the MNPs. The reaction mixture can be complemented by the addition of stabilizing agents, in order to prevent the agglomeration of the particles. Poly(N-vinyl-2-pyrrolidone)—PVP, sodium dodecyl benzyl sulfate—SDS, proteins, peptides and gums are some of the stabilizing agents used in this context. Moreover, the applied controlled conditions, including components concentration, ratios, pH of the media, selected surfactants, temperature and pressure, constitute determining factors for the final MNP properties, size and morphology [25,26].

#### 2.1.1. Co-Precipitation

Co-precipitation is a simple synthesis method for MNP production that has been reportedly used in magnetite or cobalt ferrite NP synthesis. The process uses aqueous ferrous and ferric salt solutions in a 2:1 proportion, in the presence of a base, at room temperature or higher, that can be enhanced through the application of high-pressure homogenization during the precipitation process, or under slower reaction conditions [27]. These modifications enable a more controllable MNP in terms of size, magnetic properties and crystallinity. Ge and co-workers [28] described the synthesis of paramagnetic iron oxide nanoparticles using the co-precipitation method in an aqueous solution containing ferrous and ferric chloride solutions in a 2:1 molar ratio, respectively. The precipitation was induced by the vigorous stirring of the mixture in the presence of concentrated ammonia and under nitrogen, from which resulted in a black precipitate. This was washed with deionized water, dispersed in an aqueous acidic solution (pH 3.0) and oxidized into brown iron oxide nanoparticles (IONP) by air at 90 °C [28].

#### 2.1.2. Thermal Decomposition

Thermal decomposition is an up-scalable, dilatory synthesis method applied to non-magnetic organometallic precursors in the presence of organic solvents. The organometallic precursors originate pure metal that is posteriorly oxidized at temperatures ranging from 100 to 350 °C and oxidative media, and forms metal oxides, e.g., iron oxides. Metal carbonyls or metal acetylacetonates are some of the currently mentioned metal precursors used in thermal decomposition method, and oleic acid or fatty acids are some of the selected surfactants for the mentioned technique [25]. 

#### 2.1.3. Hydrothermal Approach

Hydrothermal synthesis comprises a simple and easily up-scalable synthesis method based on the hydrolysis and dehydration of metal salts in an aqueous media under high-temperature and high-pressure conditions, that can be performed in autoclave equipment (with temperatures above 200 °C and pressure above 2000 psi). The reaction conditions enable the production of metal oxide particles (poorly soluble in the high-temperature aqueous media) and the subsequent precipitation of the MNPs. The controllable variables in this procedure involve temperature, concentration and autoclaving time, the latter influencing proportionally particle size and distribution. 

#### 2.1.4. Microemulsification

Microemulsion constitutes another synthesis method that has been performed for MNP production. This oil-in-water (o/w) method requires the use of an aqueous phase (containing the metal salts, pH regulators and possibly coating agents), an oily phase (e.g., hexane) and surfactants as stabilizing agents in the water/oil interface (e.g., sodium dodecyl sulfate—SDS, poly(N-vinyl pyrrolidone)—PVP or bis(2-ethylhexyl) sulfosuccinate—AOT). The reversed microemulsion can be also performed, being this process based on the dispersion of the nanosized stabilized aqueous phase in the oily phase. This process originates dynamic systems that may coalesce and permit reaction upon mixing, therefore providing a controlled nucleation and growth environment [29]. The reversed microemulsion can be performed under temperatures within the range of 20–80 °C. Notwithstanding, it comprises a dilatory and low scalable process.

#### 2.1.5. Polyol-Based

Polyol synthesis is applied to metal salts in a polyol media using a reduction reaction. The polyols serve as a solvent, reducing agents and stabilizers, and allow the implementation of a synthesis method without high-pressure requirements, representing an interesting process, particularly in the production of flower-shaped MNPs. This particularity is due to the reductive ability of polyols to determine the rate of structure formation and growth [30]. The polyol mixture can be composed by poly(ethylene glycol) (PEG) and diethylene glycol (DEG) or N-methyldiethanolamine (NMDEA), among others, and performed at lower pressure conditions, being the reaction conditions and the selected solvents the variables of this method that can affect the obtained NP characteristics. Curcio and co-workers [31] performed the synthesis of water-soluble iron oxide nanoflowers (IONFs) using polyol synthesis, in which ferrous and ferric chloride solutions in a 2:1 molar ratio were dissolved in a liquid mixture of DEG and NMDEA in a 1:1 volume ratio and stirred for 1 h. NaOH was used as the base, that was added to a separate polyol mixture of the same composition, mixed with the iron oxide solution and stirred for another 3 h. The thermal treatment was further performed, under an increasing temperature to 220 °C for 50 min, followed by 2.5 h stirring and room temperature cooling. The sediment (containing the obtained NPs) was subjected to magnetic separation, washing (using ethanol-ethyl acetate in a 1:1 volume ratio; a 10% nitric acid treatment at 80 °C for 45 min; washing with acetone-diethyl ether solution) and final redispersion in water [31].

#### 2.1.6. Sol-Gel

Sol-gel reaction is based on the hydrolysis and condensation reactions applied to MNP precursors, such as metal alkoxides and metal salts. This technique relies on the hydroxylation and condensation of the MNP precursors in a solution (sol) and subsequent gelation, under room temperature, which is subsequently subjected to a thermal treatment to promote the formation of the crystalline structure, control of size and shape of the desired MNP [32]. Given the different succession of the mentioned phases, this technique operates within a wide temperature range of 20–200 °C.

#### 2.1.7. Electrochemical

An electrochemical reaction is a prolonged medium-scalable synthesis technique that employs an oxidation-reduction reaction of metal salts. It constitutes an appealing and advantageous method, as it provides high-purity and strict size control of the obtained MNPs. Nonetheless, chemical synthesis can include some disadvantages, such as unwanted properties due to the presence of some components e.g., stabilizing agents or solvents [33]. The presence of solvent or surfactant residues may compromise the surface functionalization of the MNPs, their biocompatibility and desired features, and contribute for secondary effects, such as cellular toxicity [34].

#### 2.1.8. Biosynthesis

Biosynthesis comprises another possible MNP synthesis method, commonly mentioned and applied in the production of magnetic iron oxide nanoparticles (IONPs). In summary, this technique employs a red-ox reaction in vivo stemming from microbial enzyme activity and plant phytochemicals. For that, magnetotactic iron-reducing bacteria such as *Geobacter metallireducens* or *Magnetospirillium gryphiswaldense* reduce iron salts into NPs, under specific conditions (anaerobic or aerobic, depending on the selected microorganism). However, in what concerns the MNP control specifications, such as dimension and shape, further research is required, as the process does not provide strict control of these specifications.

#### 2.1.9. Summing-Up

In Table 2 are summarized the different techniques for synthesis of magnetic nanoparticles, reporting the respective benefits and limitations.

### 2.2. Hybrid MNP Synthesis

The production of hybrid constructs combining ferromagnetic elements, such as iron, nickel and cobalt, with ions, oxygen and other metals, such as noble metals, contributes to the development of new nano-sized structures with tuned properties for multimodal biomedical applications [44,45,46]. Hybrid MNPs production can be carried out through the various above-mentioned chemical synthesis methods. Co-precipitation method is an example. Other methods share relevance for the production of hybrid MNPs. In particular, chemical reduction and photoreduction will be addressed in what follows. 

The synthesis of hybrid MNPs is widely based on the deposition of noble metal (NM) NPs on the MNP core, the latter essentially represented by metal oxide MNPs, such as titanium oxide or zinc oxide NPs. The process involves an aqueous solution containing the noble metal precursor (e.g., AuCl_4_^−^), in which the MNPs are dispersed and suffer NM precursor adsorption. Subsequently, the obtained product may follow chemical reduction at mild temperatures (by the addition of reducing agents, such as ascorbic acid or sodium borohydride) or photoreduction (by light irradiation of photons of a wavelength above 300 nm) enabling the formation of NM-metal oxide MNPs [33]. The photoirradiation sources used to depend on the desired hybrid MNP and include high-pressure mercury arc (used for Au-TiO_2_ NPs), low-pressure mercury lamp (used for Ag-TiO_2_ NPs) and sunlight (used for Pt-TiO_2_ NPs).

Recent research work concerning hybrid magnetic nanoparticles for biomedical applications is predominantly focused on the development of magnetic-plasmonic heterodimers. The combined magnetic and plasmonic properties provide MRI, PTT and PDT responsive nanoconstructs, desirably with maximum surface plasmon absorption intensity and plasmon resonance absorption peak in the NIR region. Magnetic-plasmonic heterodimers can be obtained by chemical synthesis (e.g., polyol method for Ag-FeCo hybrid NPs) [47], and can be composed of metallic-metallic or metallic-nonmetallic materials, such as Fe-Au heterodimers or Fe_3_O_4_-Au heterodimers, respectively. Due to the higher stability in biological media and the predominant hydrophilicity, metallic-nonmetallic hybrid MNPs are particularly interesting as host nanoparticles, and due to the potential applications, they urge as highly promising nanotheranostic agents [48].

## 3. Physical Properties of MNPs

Size, shape, optical, electrical, thermal and magnetic properties are major constraints of MNPs suitability [49]. These properties constitute important parameters for the nanoparticle system performance in vitro-in vivo.

As mentioned, MNPs superparamagnetic behaviour is size-dependent and is preferred for biomedical applications. Thus, it is important to tune NPs features within the size range in which this property is preserved. For pharmacokinetic purposes, as nanoparticles below 10 nm are promptly cleared via renal excretion and those above 200 nm are rapidly removed from the bloodstream, the ideal dimensions for intravenous administration of a magnetic nanoparticle system are demonstrated to be included within the range of 10–100 nm [17]. On the other hand, for the intended physical performance, the increase in particle dimension within this range leads to higher magnetization saturation and an optimized response to the applied magnetic field. Therefore, a balance of these parameters is required for the design of a suitable MNP at physiological temperatures and environment [13,27].

Geometry and structure have also been reported to influence the MNPs properties and performance. The composition and architecture of the nanosystems have a significant impact on the NPs magnetization saturation (Ms) and deformability, and therefore on their suitability for imaging and drug delivery purposes [50].

MNPs exist in several formations, namely spheres, rods, disks, wires, cubes, triangles, polyhedrons, gels, cages, flower-like structures, among others [19,20,51]. Due to the reduced dimensions, they assume a high surface area to volume ratio and express high aspect ratio values. Nonetheless, this parameter depends on the NPs shape. 

A high aspect ratio may be considered a favourable feature for cellular internalization and surface functionalization, as NPs with higher aspect ratio present higher deformability, surface area and prolonged bloodstream circulation periods when compared to NPs with the same structure yet lower aspect ratio. However, for the MNP shape selection, one must have into account that the MNPs surface often contains disordered spins that do not contribute to total Ms, resulting in magnetic saturation depletion, when compared to the respective bulk material. 

Magnetic properties have been identified as shape and composition-dependent on MNPs, such as maghemite rods, ferrite cubes, cobalt disks, Ni-Fe wires and Au-MnO flower-shaped nanoparticles [19]. The correlation between these two parameters is due to the shape anisotropy influence on the orientation of the magnetic moments of the particles. The shape-induced magnetic anisotropy interferes with magnetization reversal due to thermal fluctuations and affects the heating efficiency of NPs [52]. On this basis, higher heating capacity and magnetization saturation values have been associated and demonstrated in cubic ferrite MNPs (of lower surface anisotropy) when compared to spherical MNPs of the same size [19,27,53].

Optical characteristics are the tunable size and geometry-dependent features in MNPs. For biomedical application purposes, these MNPs represent great relevance for multimodal imaging techniques, especially in MRI-optical imaging. Optical properties in noble metals, such as gold (Au) and silver (Ag) [54] comprise an interesting physical effect, given the strong resonances stemming from electron transitions in these monovalent metals, that plays an important role in the optical contrast enhancement.

Attempting a more efficient combination of magnetic and optical characteristics, hybrid noble metal-metal oxide NPs offer viable options for theranostic MNPs, e.g., gold-coated or silver-coated iron oxides [20,33].

Desirable optical properties can also be attained through the incorporation of fluorophores, as demonstrated by Foy et al. [55], who have developed nanoparticle systems with the surface conjugated with a near infra-red (NIR) hydrophobic dye of intense fluorescence (e.g., SDB5700 and SDA6825 commercially available dyes) [55].

Electrical or charge properties are related to the surface charge of the obtained MNPs. This is a determinant factor for the magnetic nanoparticle performance, in terms of imaging and drug delivery purposes, as it influences targeting, as well as cellular uptake and intercellular localization [56]. Surface charge is mostly tuned with the coating of the MNPs (e.g., smart polymer coating) and is measurable by determination of the zeta potential using electrophoretic light scattering (ELS) [28]. [57]. As an example, Weidner et al. [58] prepared pure spinel structured MNPs (not coated superparamagnetic iron oxide NPs) of a 9.6 nm mean size, Ms of 68.2 Am^2^/kg, coercive field (Hc) inferior to 0.2 kA/m and relative remanence (Mr/Ms) of 0.005 at room temperature, which presented a positively charged surface slightly above +30 mV. The study also involved the analysis of the surface charge influence of three different coatings in the superparamagnetic iron oxide NPs, in which MNPs acquired the components charge when coated with the neutral agent dextran (DEX) and presenting a slightly positive surface charge, the negatively charged carboxymethyl-dextran (CMD) acquiring a surface charge lower than −30 mV, and positively charged diethyldiaminoethyl-dextran (DEAE) MNPs with surface charges above +60 mV [58].

## 4. Characterization of MNPs

The characterization of MNPs requires the use of several methods and pieces of equipment, in order to determine all the inherent properties of the product and its suitability for the intended biomedical use.

The dispersity, morphology and structure of the MNPs can be determined by high-resolution transmission electron microscopy (HR-TEM), energy dispersed spectroscopy (EDS) and powder X-ray diffraction (XRD); and the hydrodynamic size using dynamic light scattering (DLS), also known as photon correlation spectroscopy (PCS) [57]. Accordingly, MNPs are dispersed in a solvent and subjected to sonication before equipment software analysis. PCS or DLS uses particle Brownian motion to extract the hydrodynamic radius, size distribution and colloidal stability of the MNPs. Information in what pertains to surface coating, as well as MNPs stability under biological conditions can also be inferred [59].

The estimation of the magnetic core size of iron oxide MNPs is based on the Chantrell method [58,60]. This method enables the determination of the median particle diameter (MPD) and respective standard deviation (SD) of the entire sample volume, calculated from the room-temperature magnetization curve, providing more sensitive and reliable results when compared with TEM.

TEM and high-resolution scanning electron microscopy (HR-SEM) imaging enable the analysis of shape, size and distribution of MNPs, providing information complementary to that obtained from XRD. HR-TEM also qualifies to the analysis of the MNPs crystallinity. The high-resolution imaging enables the observation of the growth direction, planes and lattice spacing between the latter, being considered an alternative method for the analysis of single crystal formation. 

Structural analysis using X-ray diffraction requires the dispersion of MNP powder onto a silicon sample holder for further scan analysis at a determined scan velocity (e.g., 10° to 90° scan of 0.02°/s for magnetite NPs, or a scan mode of 0.06°/s within a 20°–80° range for nickel ferrite NiFe_2_O_4_ MNPs) under CuKα applied radiation. The process generates a spectrum comparable to international databases and facilitates the detection of impurities [61]. 

Functional groups and chemical bonds of the MNPs coating materials can be confirmed using Fourier transform-infrared spectroscopy (FT-IR). FT-IR spectrum provides information on the vibrational properties of MNPs. For example, the bands that are assigned to the vibration of ions in crystal lattices report on the composition of the MNPs.

Raman spectroscopy can also be used as a complementary spectroscopic analysis technique, to provide relevant chemical and structural information on the MNP surface molecules [62]. 

In addition, information relative to the study of the magnetic core characteristics can be obtained using Mössbauer spectroscopy. The technique has been reportedly used, e.g., in iron oxide nanoparticles, and is particularly interesting for the discrimination of the valence of the iron ions present in the MNPs. Thus, Mössbauer spectroscopy can be useful to distinguish magnetite versus maghemite arrangements [63].

The magnetic properties and performance can be determined using a SQUID-VSM magnetometer or a vibrating sample magnetometer (VSM) at a determined temperature and applied magnetic field. This equipment provides the sample analysis and generates a magnetic hysteresis curve, from which the magnetic behaviour, saturation magnetization and coercivity of the sample can be inferred [64]. Furthermore, information on the orbital magnetic moment and spin of the atoms, retrieved from X-ray magnetic circular dichroism (XMCD), can complement the MNPs characterization, using. For this purpose, XMCD makes use of the material’s polarization-dependent light absorption, which varies for ferromagnetic and paramagnetic particles in function of helicity of the circular polarization vector, whether it is parallel or antiparallel to the externally applied magnetic field. The XMCD spectra results from the difference between the X-ray absorption spectroscopy (XAS) spectra of opposite circular polarization, which provides an electronic and magnetic fingerprint of the MNP in the study [65,66].

The combination of the above-mentioned analysis methods is crucial for the MNPs characterization, assessment of suitability for the intended use and performance in the physiological conditions.

## 5. Protection/Stabilization

Within the MNPs, metallic iron nanoparticles are commonly associated with biotoxicity and chemical instability due to the high oxidation susceptibility and propensity towards aggregation. Surface coating methods with inorganic materials such as gold, silver, silica or Co_3_O_4_ can overcome iron MNPs limitations and enable the protection and stabilization of these nanoparticles [17,43,67]. Proper protection of the MNPs will ensure the stability of the nanoparticles, avoid magnetic properties loss (compromised by NP oxidation) and prevent NP aggregation. Among the possible surface coating materials, silica represents one of the most promising constituents, providing a biocompatible, non-toxic, chemically inert coating with a silanol terminus that enables the attachment of specific ligands to the MNPs surface through covalent bonds [68]. These methods are of high relevance, as they can be tuned to provide functional groups for biomolecule conjugation, enhanced control over drug release and an NP extended half-life [18]. 

Coating materials can be divided into inorganic or organic. As previously mentioned, inorganic materials used in MNP coating include noble metals, silica or metal oxides. On the other hand, organic coating materials used to produce MNPs include dextran, starch, poly(ethylene glycol), poly (D,L-lactide), polyethylenimine (PEI), among others. For example, MNPs coated with noble metals, such as gold, demonstrated effective application as contrast agents in multimodal imaging techniques, including MRI, ultrasound (US) and computerized tomography (CT). Magnetic iron oxide nanosystems coated with cobalt tetraoxide (Co_3_O_4_) originated a core-shell hetero-nanocomposite that provided MNP stabilization, more controllable shell thickness and enhanced magnetic behaviour [43]. In turn, magnetic nanosystems of reduced size, sugar-coated (glyconanoparticles) and MNPs presenting negative surface charge have demonstrated to exhibit biocompatibility, and prolonged blood circulation time [69]. 

Recent efforts have been focused on improving the thermodynamic stability of the MNPs [70]. Besides the use of thiol group molecules, monomeric stabilizers such as carboxylates, phosphates and phosphonates have been explored, as they exert a strong interaction with the MNP surface by forming combined covalent and physisorptive bonds [71,72]. The binding efficiency of these monomers has raised great interest, for the development of thin-coated MNPs using phosphate, phosphonate, carboxylate and thiol groups derived molecules as coating agents. As an example, a comparative study on the stability of modified PEG-coated MNPs, in which monophosphonate and triphosphate were used as PEG anchoring groups, demonstrated higher MNP stabilization and lower phosphate salt interaction under physiological conditions with the triphosphate modified PEG-coated MNPs [25].

Coating materials ascribe several effects to MNPs. These include protection, stability, biocompatibility and functional groups for biomolecule attachment, and are intended to produce a minimum impact on the MNPs magnetic properties, except when providing contrast enhancement. Thus, the surface coating is an important step in MNPs design and production [24]. 

## 6. Functionalization

The functionalization of MNPs served two main purposes in the past: enhancing biocompatibility and avoiding NP aggregation. However, with technological development, it tended to evolve towards the use of functional moieties with a broader diversity of functions, in which the focus is not only interface properties engineering (as wetting and adhesion) but also enhancing intracellular trafficking and overcoming cell membrane barriers. Electro-sensitive polymers constitute an example of MNP functionalization, using energy from an electrical field and convert it into mechanical energy that can be observed in the form of shrinking, bending or swelling behaviour [73]. Polymers such as chitosan, alginate and poly(dimethylsiloxane) have been used for the preparation of electro-sensitive materials, the latter exhibiting stimuli response by gel bending behaviour in the presence of colloidal silica oxide particles [74]. These are essentially electrochemical sensitive polymers, which are pH-sensitive as well, and consequently, present some interesting properties for drug delivery systems. However, the functionality of these polymers has been proven in the absence of electrolytes, which cannot warrant the polymer function in physiological environments [75].

Also, photo-functional moieties have been used, serving as photosensitizing agents or fluorescent dyes for imaging purposes [68]. Other components, such as chemotherapeutics for single/multidrug therapy and biotherapeutics for siRNA or peptide therapy may be combined with the MNPs. Briefly, these may employ surface polymer functionalization, alone or in combination with therapeutic and contrast agents for theranostic systems, amphiphilic block copolymer functionalization, micelle incorporation of MNPs and biomolecule functionalization [76].

### 6.1. Polymer Functionalization

MNP functionalization may be achieved using polymer functionalization. This process has demonstrated a positive impact on colloidal stability, pharmacokinetics, biodistribution and biodegradability, and more recently on triggered release by employing smart polymer coated MNPs. Smart polymers comprise polymers that assign specific properties to the MNPs, such as pH-, electro-, thermo-, and photo-sensitive response, to endogenous or exogenous stimuli [43].

pH-sensitive polymers are, in general, polyelectrolytes with included weak acidic or basic groups, that change dimensions in response to media pH, by accepting or releasing protons. Their properties depend on the present functional groups (e.g., -COO-; -NH_2_; -OH; -NH-; -NHCO-; -O-). Polymers containing acidic groups, such as polyacrylic acid, tend to expand in basic media; as for polymers containing basic groups, such as -NH_2_, the process occurs in acidic pH. Tertiary amines and imidazole groups have been exploited due to their protonation capacity and resulting electrostatic repulsion in the function of the media pH, that were revealed to be favourable for drug release [77]. Cancerous cells are known to have increased proliferation and metabolic rate, consequently exhibiting lower values of intercellular pH. Accordingly, magnetic nanoparticles can be used to deliver therapeutic drugs, in response to the endogenous stimuli, that is, the slightly acidic pH media [78]. 

A recent example [79] consisted of the synthesis of camptothecin loaded pH-sensitive MNPs containing acetylated β-cyclodextrin. The nanosystem was prepared by emulsification of iron oxide NPs into a previously acetylated β-cyclodextrin-dichloromethane solution containing camptothecin dissolved in anhydrous dimethyl sulfoxide, through sonication. The resulting preparation was further emulsified into a gelatin solution, for dichloromethane evaporation, and the final MNPs collected from the w/o/w after centrifugation were washed and lyophilized. This method permitted the synthesis of 1:10 ratio core-polymer MNPs containing a hydrodynamic size of 234 nm, polydispersity index of 0.164 and zeta potential of −13.8 mV, with pH response at a value of 5.5 with acetal degradation and reversible magnetization. The magnetic saturation values were superior in acidic media (pH = 5.5), suggesting that polymer coating reduces MNPs magnetization. Furthermore, the magnetic nanosystem exhibited a 75% drug loading capacity and successful drug release in acidic media within 30 min, when compared to the 20-h drug release for pH 7.4, which indicates a promising dual-stimuli responsive platform for drug delivery in cancer treatment [79].

A fine-tuning step on the use of smart polymers relies on the development of combined multi-theranostic nanosystems. In this regard, the synthesis of digenite (copper sulfide Cu_9_S_5_) silica-coated nanoparticles covered with iron oxide nanoparticles, PEG modification and doxorubicin payload was described [80]. These magnetic nanocomposites integrate a chemo-photothermal therapy functionality, due to Cu_9_S_5_ capability to convert NIR light to thermal energy and due to the thermally enhanced drug release upon NIR light incidence of 980 nm, combined with polymer pH-responsive drug release and iron oxide MR imaging applications, representing the interest in MRI assisted and monitored cancer therapy [80].

### 6.2. Amphiphilic Block Copolymer Functionalization

Technological development has further evolved towards functionalization of MNPs using amphiphilic block copolymers. This process endows more functional groups to the magnetic NPs, entailing a higher interest related to the multifunctionality and biological applicability of the MNPs in cancer nanotheranostics. An example of this approach consisted on the use of hydrophobic iron oxide nanocrystals in chloroform, dispersed into an aqueous phase containing cetyltrimethylammonium bromide (CTAB) stabilizing surfactant, and further coated with silica and surface modified with an amine group and PEG [81]. Amine functional groups were added to the obtained mesoporous silica-coated IONPs, that were then submitted to surface modification with PEG, for prevention of non-specific surface protein adsorption.

This method enabled the production of 15 nm core size SPIONs, enlarged to 53 nm after silica coating and with mesopores of 2.6 nm. PEG-modified MNPs presented a hydrodynamic size of 97 nm and no aggregation phenomena were observed [81]. These nanoplatforms were further tested for imaging applications by MRI, which demonstrated a darker T2 relaxation signal with increasing MNP concentration, and drug delivery applications using doxorubicin-loaded MNPs in a human breast cancer cell line (SK-BR-3 cells). Nanoparticles that did not contain the chemotherapeutic drug exhibited a non-cytotoxic behaviour, while the doxorubicin-loaded nanosystem demonstrated a proportional and MNP concentration-dependent increased cytotoxicity. Furthermore, the synthesized magnetic nanoparticles demonstrated successful cellular internalization to the cytoplasm with preservation of the NPs’ structure and theranostic potential [81]. Another related approach is the micelle synthesis and MNP incorporation. Poly (ε-caprolactone)-b-poly(glycerol monomethacrylate) and poly (ε-caprolactone)-b-poly(olygo(ethylene glycol) monomethyl ether methacrylate-co-folic acid) are copolymers that have been used to obtain functionalized superparamagnetic NPs and drug-loaded hybrid amphiphilic block copolymer micelles, functionalized with folate moieties. This method was employed in a study performed by Hu and colleagues [82], in which the copolymers were synthesized, subjected to comicellization, followed by the insertion of a chemotherapeutic drug (paclitaxel) into the micellar cores and the SPIONs loading into the respective hydrophilic coronas [82]. The poly (ε-caprolactone) portion corresponds to the hydrophobic core that enables the effective drug encapsulation and the poly(glycerol monomethacrylate) polymer portion comprises 1,2-diol moieties with high affinity towards the MNPs surface iron atoms, constituting an interesting system for targeted chemotherapeutic drug, sustained release and MRI imaging contrast.

### 6.3. Biomolecule Functionalization

Biomolecule functionalization is a step forward towards increasing MNP biocompatibility, using biomolecules, such as enzymes, antibodies, biotin, polypeptides, proteins, bovine/human serum albumin and avidin bonded to the MNPs surface. For example, carboxylated silica surface coated MNPs can be covalently bonded to amine groups (through an amide bond), present in the previously mentioned biomolecules. The synthesis of carboxylic and amine-functionalized silica-coated iron oxide nanoparticles was performed by Jang and colleagues [68]. Superparamagnetic IONPs were produced using the co-precipitation technique in the presence of ammonium hydroxide with subsequent heat treatment and washing. Sequentially, 100 mg of the obtained MNPs were dispersed in a 4:1 anhydrous ethanol/de-ionized water mixture with ammonium hydroxide for pH adjustment (pH = 9) and 200 µL of tetraethyl orthosilicate (TEOS) of high purity. The mixture was submitted to sonication and agitation for a 4 h period under inert argon gas and the formed precipitate of silica coated MNPs was washed and stored in an ethanol solution. For the amine immobilization as the MNPs’ functional group, 2 mL of the silica-coated MNPs were washed and dispersed in toluene and N,N-dimethylformamide (DMF) in a 1:1.5 ratio (8 mL to 12 mL). One millilitre of aminopropyl triethoxysiliane (APTES) was then added dropwise using a needle syringe and subsequently stirred at room temperature for 24 h. This process is based on the TEOS and the APTES’ triethoxy-silicon group hydrolysis and condensation reaction, to form an inorganic network in the MNPs’ surface, endowing these NPs with an amine functional group [83]. Finally, the amine-functionalized silica-coated MNPs were washed and stored in toluene. Furthermore, for carboxylic functionalization, glutaric anhydride was added to the previous product under stirring. Accordingly, the acid anhydride and the MNP’s surface amine group reaction forms an amide bond and attributes a free functional carboxyl-terminal group to the MNP. The final MNPs-bearing a carboxylic functional group were washed and dispersed in toluene [68]. 

The application of biomolecule functionalization to MNPs comprises a different strategy for enhancing stability in the diverse conditions of the biological media (e.g., pH and temperature) enabling the design of non-enzymatic biosensors for biomarker detection in cancer diagnosis [69]. Platforms of this nature have further application, for example, by conjugation of the Glut antibody with the MNPs in the detection of glucose transporter, protein Glut-1, which have been demonstrated to be significantly involved in glucose transport in cancer cells [69].

## 7. Nanotoxicology

Due to the diversity of configurations, materials and properties of MNPs, magnetic nanoparticles can be considered as inherently complex systems. Thus, the determination or estimation of the associated cytotoxicity is a difficult process that requires detailed and directed guidance in terms of quality, toxicology and monitoring of MNPs. The exposure to MNPs has been predominantly associated with toxicity effects, such as oxidative stress, mitochondrial and cell cycle impairment, oxidative DNA damage and protein denaturation. Therefore, among the toxicological parameters, it is important to address the carcinogenic potential, genotoxicity, tendency to aggregation, immunotoxicity and reproductive toxicity of each MNP [84]. 

For example, the formation of reactive oxygen species (ROS) has been associated with IONPs toxicity and, thus, one possible approach for the respective prediction would be intracellular ROS measurement. The determination of the MNPs effect upon ROS production in cells can be measured by fluorescence microscopy using a DCFDA (2′,7′-dichlorofluorescein diacetate) fluorescent probe and quantitatively confirmed using flow cytometric analysis. Accordingly, the MNPs loaded with the reduced non-fluorescent form (2′,7′-dichlorohydrofluorescein diacetate or H2DCFDA) enter the macrophages and neutrophil cells and the H2DCFDA is converted into the fluorescent agent upon acetate group cleavage by intracellular oxidative processes. The intense fluorescent signal is then detected by flow cytometry [85,86].

Furthermore, ROS overproduction can be the origin of mitochondrial impairment, compromising the mitochondrial membrane potential. 3,3′-Dihexyloxacarbocyanine iodide (DiOC6) green-fluorescent lipophilic staining followed by cytometric flow analysis enables the measurement of the MNP exposure effects on the membrane potential, resulting in augmented green fluorescence intensity with the increase of membrane potential [87].

Cytotoxicity can be evaluated using Hoechst 33342 and propidium iodide (PI) staining for the confirmation of cell death, as performed by Feng and colleagues [87]. In viable cells, Hoechst 33342 stains the chromatin with a blue colour that can be observed in fluorescent microscopic images, while the PI (that results in red-stained nuclei) is only permeable to non-viable cells, enabling the distinction and confirmation of the MNP induced cytotoxicity, upon different concentrations. 

Cell membrane integrity is passible of determination through the LDH (lactate dehydrogenase) enzyme assay, based on the correlation of the increasing LDH release with the increase of lysed cells. 

Hemolysis determination is a facile method for the evaluation of the hemolytic properties of MNPs. This method can be performed by measuring the spectrophotometric absorbance of the sample at 540 nm based on the haemoglobin level, taking into consideration a positive and a negative control [88].

Annexin V and PI staining constitute an apoptosis discrimination method. This process is based on the annexin affinity to binding to phosphatidylserine (PS). PS is present in the inner portion of the plasma membrane under normal conditions, however, under early apoptosis is translocated to the external leaflet of the cell membrane, serving as a marker for phagocytosis. Under these conditions, is it possible to bind fluorochrome-conjugated annexin V to PS and proceed to the analysis by flow cytometry, while viability dyes such as PI do not stain these apoptotic cells. Thus, annexin V serves as an apoptosis marker when the cell membrane integrity is assured. In late apoptosis, the cell membrane integrity is compromised, therefore allowing the intercellular binding of annexin V and exhibiting the PI red staining. This method enables the distinction between apoptotic and necrotic cells [89].

MNP induced cell cycle impairment can be evaluated by flow cytometric analysis using propidium iodide. PI is a DNA-binding dye that binds proportionally to the DNA present in the cell, the latter being dependent on the cell cycle phase. Thus, the S phase and G2 phase cells present quantitatively more DNA when compared to G1 phase cells and, therefore, exhibit more intense fluorescence. As mentioned above, PI does not cross intact cell membranes. As such, membrane permeability in both cell membrane and nuclear membrane is required, which is induced by detergents such as Triton X-100 in 0.1% and fixation using, e.g., alcohol. This method provides the estimation of cell count in each cell cycle phase, using an algorithm to fit Gaussian curves to the obtained results, using the flow cytometry software. 

Other tests, including the determination of proteins involved in cell mechanisms (e.g., anti-apoptotic Bcl-2 and pro-apoptotic Bax), cell cycle regulation (e.g., cyclin D) and autophagy markers (e.g., LC3B-II) can be determined by western blot. The formation of autophagosomes can be identified using TEM imaging technique [87].

Regarding DNA damage, genotoxicity analysis can be conducted by the single-cell electrophoresis Comet assay. Briefly, and adapted from the described method in the protocol of Singh et al. [90] and Tice et al. [91], cells exposed to varying concentrations of MNPs are stained with a solution of ethidium bromide, subjected to electrophoresis and readily analyzed by fluorescence microscopy, in which the increasing migration of the DNA suggests increasing DNA damage [92]. Könczöl et al. [92] performed this assay using magnetite NPs, having demonstrated dose-dependent magnetite induced DNA migration and, consequently, DNA damage. 

Additionally, body weight, haematology and blood biochemistry monitoring can provide important information regarding in vivo toxicity, as an analysis method of MNPs influence in the organism.

## 8. Biophysical Mechanism

The progressive evolution and development of MNPs have raised concerns regarding the nanoplatforms biological behaviour, distribution and clearance in a living organism. The same performance in a biological environment is influenced and determined by the NP design and final intrinsic physicochemical properties, as well as by the variation on the media conditions and composition. All these factors challenge the determination of MNP biophysical and chemical mechanism [87]. 

Note that these nanosystems are mostly administered intravenously, in which there is an expected contact with plasma proteins. This initial environment subjects MNPs to possible surface protein adsorption, known as opsonization, followed by macrophage recognition and uptake, resulting in rapid MNP elimination from the blood circulation. As previously mentioned, this limitation can be minimized through the insertion of coating materials on the MNPs surface. 

Particle size and polydispersity [25] also condition the MNP circulation period, reproducibility and biological distribution, being the homogenous and reduced size superparamagnetic NP (of above 10 nm), subjected to slower opsonization and clearance processes, and able to evade macrophage phagocytosis (when under 100 nm).

The biodistribution of magnetic nanoparticles in an in vivo system can be analyzed by transition metal detection using inductively coupled plasma-mass spectrometry (ICP-MS) in a tissue sample. This method enables the detection of reduced metal concentrations, combining argon gas plasma and a mass spectrometer. The sample is diluted, subjected to the digestion of organic solvents with 37% nitric acid, then exposed to the inductive coupling plasma (ICP), therefore ionized and separated by mass/charge ratio, under an applied electric field. These data result in a mass spectrum that displays the intensity of the metal content signal (expressed in mV) in the function of the mass/charge ratio, the former being proportional to the element concentration in the sample. The excess metal in the analyzed sample suggests the accumulation of MNPs in the respective organ [93]. Feng and colleagues [87] demonstrated the IONP propensity of accumulation in liver and spleen, and to a lower extent, in the lungs, heart and kidney, in 24-h tissue samples. A large accumulation in the renal system may suggest rapid clearance. The same analysis was performed at the 24th hour after MNP injection, using a serum sample that reported practically undetected iron concentration, therefore indicating a rapid blood clearance. For the clearance and degradation analysis throughout time, the Prussian blue iron staining method was performed in dissected liver and spleen tissue samples of the 6th hour and second week after the MNP injection, in which positively charged (+29.28 mV) PEI-coated MNPs displayed progressively reducing iron staining, while approximately neutral (−0.52 mV) PEGylated MNPs maintained significant staining in both samples. The results suggest that the MNPs surface charge has a significant impact on MNP internalization and degradation, with the positively charged NPs revealing more affinity to phospholipid membranes, resulting in significant internalization and degradation, and consequently, in the reduction of iron staining, when compared to the approximately neutral MNPs [94].

Furthermore, in order to analyze the MNPs intracellular distribution in a biological system, TEM imaging combined with energy-dispersive X-ray spectroscopy (EDX) can be used. The structural cell details can be observed at the sub-organelle level, providing a viable method for the analysis of MNP intercellular localization in tissue samples [95]. 

Accordingly, the previously mentioned PEI-coated MNPs demonstrated cell membrane accumulation and endocytic internalization in TEM imaging, while PEG-coated MNPs presented only endocytic internalization, and none of the prementioned MNPs were present in the cell nucleus. 

## 9. Medical Applications

MNPs have demonstrated potentialities for diagnosis and therapeutics. In the context of diagnosis, these nanosized ensembles can be designed to perform as contrast agents in MRI, Positron Emission Tomography (PET), Computed Tomography (CT), Single Photon Emission Tomography (SPECT), Photoacoustic Imaging (PAI) and Surface Enhanced Raman Spectroscopy (SERS). On the other hand, MNPs for therapeutic applications may serve as drug delivery agents, gene delivery agents and/or thermoablation agents tuned for magnetic hyperthermia, Photothermal Therapy (PTT) and/or Photodynamic Therapy (PDT) [69]. 

Magnetic resonance imaging is a non-invasive diagnostic tool that uses a magnetic field and radiofrequency electromagnetic pulse waves to provide a high spatial resolution of internal structures of the body. The imaging technique enables the detection of the variation on the direction of the rotation axis of the protons of water molecules, resulting in soft tissue contrast imaging and, as such, for contrast enhancement, requires the use of agents that affect the protons magnetic relaxation. Contrast enhancement is promoted by T1 or T2 contrast agents. T1 contrast agents, such as gadolinium, Gd (III), reduce longitudinal relaxation time and generate a brighter signal. T2 contrast agents, such as superparamagnetic nanoparticles (e.g., SPIONs), generate local magnetic fields that interfere and reduce the transverse relaxation time, resulting in a darker signal. Spherical nanocomposites combining both contrast agents, dextran-coated SPIONs with Gd ions inserted in the external MNP structure, have been synthesized by co-precipitation with Gd (III) nitrate for dual-modal imaging [19]. The Gd-doped system presented a nano-sized platform not only for MRI but also for hyperthermia therapy, making use of the relaxation energy loss in the form of heat to induce cell damage, that is intended to address the tumour tissue [19]. Further investigation would be required to produce MNPs with targeting moieties to direct the nanotheranostic agent to the target site and avoid negative interference and damage of healthy body cells and tissues. CT comprises an X-ray-based measurements technique that provides a cross-sectional X-ray imaging construct of the determined area of the body. It is a faster, less expensive method and provides images of tissues, organs and skeletal structure when compared to MRI, however, the latter represents a more advantageous method, due to the absence of the exposure to ionizing radiations. Alternatively, gold nanorods and spherical nanoparticles have been studied as contrast agents for the mentioned technique, due to the significant X-ray attenuation (with relevant X-ray scattering and absorption phenomena), when compared to normal tissue; higher proton attenuation coefficients, compared to iodine (the most commonly used contrast agent); and reduced toxicity. The particularity of these NPs is the potential association with IONPs for dual imaging integrating MRI and CT techniques [96].

PET is a highly sensitive molecular imaging that detects the radiation activity of positron emission radioisotopes, such as copper ^64^Cu^2+^, gallium ^67/68^Ga^3+^, indium ^111^In^3+^, fluorine ^18^F^−^, among others. MNPs designed for PET imaging require the formation of a complex with the selected radioisotope. Thus, macrocyclic chelating agents, such as 1,4,7,10-tetraazacyclododecane-1,4,7,10-tetracetic acid (DOTA) are commonly used, since they tend to originate highly stable complexes, preventing transchelation events [97]. The obtained MNP loaded with the active radioisotope undergoes biodistribution in the body, accumulating in the highly chemically active areas/tissues and resulting in a bright signal that is detected by the equipment. 

Similar to PET scan, single photon emission tomography (SPECT) exhibits high molecular sensitivity and reduced spatial resolution. Thus, the combination of either of the prementioned methods with a spatial resolution imaging technique would address and bridge their limitations and provide an anatomical and functional tool for diagnostic purposes. 

Photoacoustic Imaging (PAI) is a hybrid imaging modality that combines optical and ultrasound imaging. The ultrasound imaging provides a high spatial resolution and the optical imaging delivers high contrast based on the tissue optical absorption [98]. The combination of both imaging techniques offers more information, greater specificity and penetration depth. Furthermore, PAI may provide a non-invasive method for the determination of the temperature distribution in tissues, as a relevant approach for monitoring of photothermal cancer therapies with highly laser light-absorbing contrast agents, such as gold-coated MNPs [98].

SERS comprises a vibrational spectroscopy technique of molecular signal amplification, with potential for in vivo imaging [99]. SERS uses the inelastic light scattering of photons that interact with matter, such as MNPs, to retrieve information about the surface characteristics and components. However, the technique is restricted to materials with high SERS properties, such as silver, gold and copper, and requires the preservation of these properties for in vivo cancer imaging, which has been reported as limited, when incorporated in theranostic nanoplatforms [100,101]. Li et al. [100] describe the design and synthesis of gold shell-core IONP (nanoflower-shaped NPs) with a rough surface for enhanced light scattering phenomena. The designed MNPs provided multimodal imaging application- SERS sensitivity, precise PA imaging and defined spatial resolution in MRI- and photothermal therapy performance, having expressed significantly elevated temperatures (dose-dependent temperature elevations of around 17 and 49 °C) for tumour ablation.

Regarding the therapeutic approaches, hyperthermia corresponds to one of the most explored. For the thermoablation procedure, various techniques have been implemented, such as laser, microwaves and ionizing radiation. However, the induced interference in the genetic material and low therapy selectivity have been identified as important side effects that can lead to healthy cell damage. Thus, the MNP-induced magnetic hyperthermia (MHT) provides an externally controllable local heating directed to a determined region of the body, which reduces the risk of damage of healthy tissues, in comparison to the other techniques mentioned above [13]. 

PTT is also a thermoablation technique that uses an infrared laser to activate light-absorbing MNPs, resulting in a higher heating capacity per nanosystem, when compared to magnetic hyperthermia. However, the incident infrared light penetration capacity is regarded as the most relevant of the technique limitations. Thus, the combination of both thermal therapies in a single MNP constitutes a beneficial merged therapeutic tool, that not only responds to the externally applied magnetic field, but also to the incident light [31]. 

A tri-modal therapy can also be integrated in a single nanosystem, by MNP tuning for MHT, PTT and Photodynamic Therapy (PDT). The latter technique involves the incidence of selective wavelength radiation that will induce the excitation state of a photosensitizer, which will transfer the energy to the surrounding oxygen molecules and originate the production of ROS and consequent cell death. A nanohybrid system composed of a multicore IONPs and a copper sulfide shell was produced using the polyol method and described as an optimized nanotheranostics platform with MRI and MHT responsive core, and PTT and PDT responsive shell. The integrated tri-modal thermal therapy nanoplatform conjugates an interesting system of cumulative heating capacity, that can provide a beneficial low dose approach in cancer nanotheranostics [31].

Magnetic nanoparticles can also serve as drug delivery systems, providing an extended surface area for drug loading and an optimized bioavailability, associated with lower drug administration doses and increased tissue selectivity [84]. Furthermore, the incorporation of chemotherapeutic agents in these nano-sized platforms can be achieved by the previous coating with agents that provide coupling points for conjugation, complexation or encapsulation of the selected drug. The MNPs loading capacity for anti-cancer drugs has been studied, being doxorubicin (DOX) the most widely used chemotherapeutic agent, with described application in various cancers, such as breast, ovarian, lung, thyroid cancer, and others. Doxorubicin exerts the chemotherapeutic function by intercalation in the DNA, disruption of the DNA repairing process mediated by topoisomerase-II and induction of cell damage by the production of ROS [102]. 

Gene delivery is a more recent therapeutic approach with potential for application in cancer treatment. MNPs conjugated with siRNA molecules are strong candidates for the development of multifunctional nanoplatforms that interfere with the protein translational processes in the cytoplasm and inhibit gene expression in tumor cells, see Figure 2. Due to the MNP properties, these nano-sized constructs display imaging and therapeutic applications, representing interesting theranostic agents [103]. 

To date, an increasing variety of magnetic nanoplatforms optimized for cancer theranostics have been developed and some relevant works are mentioned in Table 3.

Cancer nanotheranostic field is currently in expansion and the technological evolution requires continuous development of safer, more specific, sensitive and cost-effective strategies, in order to meet the required efficiency and efficacy for the MNPs’ performance [77].

## 10. Future Challenges

The technological progress regarding synthesis, characterization, biocompatibility and cytotoxicity of MNPs is evident, revealing positive advances in terms of quality, stability and applicability of these nano-sized platforms for imaging, diagnosis and treatment purposes. Despite the potential benefits of MNPs and their favourable outcomes for biomedical and theranostic applications, some fundamental points still require the researchers’ attention [113].

Further research addressing the biophysical mechanism, investigation of the different magnetic-based nanoparticles (besides IONPs), exploration of novel constructs and the establishment of detailed nanotoxicity guidelines are some interesting topics that require focus in the near future. Despite some MNPs have already entered into clinical programs, the focus is predominantly directed to either imaging or therapeutic applications and not focused on magnetic nanotheranostic systems (one example being Feraheme®, ferumoxytol-MNPs, that provide magnetic resonance imaging systems with no features or approved indications for cancer treatment, or chemotherapeutic systems with no explored imaging applications), for which additional studies are still demanding, envisioning the translation into the clinical practice [69]. In Table 4, several clinical trials concerning magnetic nanoparticles for diagnosis and/or cancer treatment are mentioned.

## 11. Conclusions

The universe of magnetic nanoparticles has been extensively studied in the past years, and most recently oriented for nanotheranostics application. The reduced size, controllable intrinsic physicochemical properties, externally applied magnetic field response and multi-surface functionalization constitute appealing features that make MNPs advantageous nanoscale systems for targeting, imaging and drug delivery, in the theranostic field. The main objectives and concerns in this field rely on the development of nanosystems that gather stability in biological environments, controlled drug release, high sensitivity for diagnosis, and reduced toxicity. Thus, the development of specific guidance and classification systems is an important step to most efficiently characterize and parametrize MNPs. The increasing investment in research, in European current projects, is a positive and crucial aspect for further technological development in the nanotheranostic field concerning magnetic nanoparticles. Cancer nanotheranostics is a growing area with great potential concerning biomedical applications and opened for innovation and development of new strategies to address the diagnosis and therapeutics requirements.

## Figures and Tables

**Figure 1 materials-13-00266-f001:**
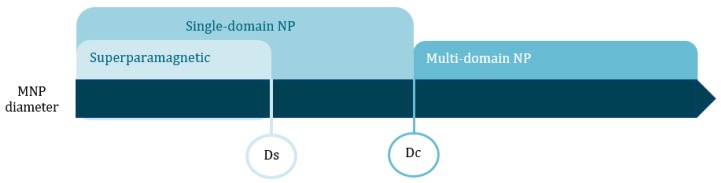
Correlation between MNP (magnetic nanoparticle) size and magnetic domain. Key: Ds stands for superparamagnetism diameter threshold and Dc stands for critical diameter threshold.

**Figure 2 materials-13-00266-f002:**
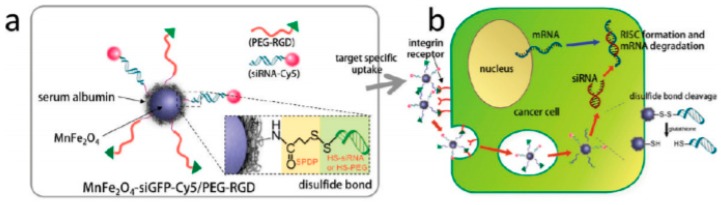
(**a**) All-in-one nanoparticles of MnFe2O4siGFPCy5/PEG-RGD for theranostic purposes. (**b**) Schematic illustration of intracellular processes of MnFe2O4siGFPCy5/PEGRGD nanoparticles, from target-specific uptake to mRNA degradation. Reprinted with permission from [103].

**Table 1 materials-13-00266-t001:** Superparamagnetism (Ds) and critical (Dc) size thresholds for MNPs.

Material	Ds (nm)	Dc (nm)
FePt	4	55
FeCo	16	51
Fe_3_O_4_	25	82
γ-Fe_2_O_3_	30	90
Co	10	80
CoPt	3	57
Co-Fe_2_O_4_	10	100
Ni	30	85

**Table 2 materials-13-00266-t002:** Benefits and limitations of the MNP synthesis methods.

MNP Synthesis Methods	Advantages	Disadvantages	References
Mechanical attrition	Simple; inexpensive equipment; adequate for scale-up.	Contamination from the materials in the media and/or atmosphere; difficulty to consolidate the powder core without coarsening the crystalline structure.	[35,36]
Thermal quenching	Up-scalable process; favorable composition control.	Elevated temperatures required; large size distribution; lack of homogeneity in microstructure.	[37]
Pyrolysis	Reduced reaction times; high purity.	High-pressure and temperature conditions; gas as adsorbent and carrier; large size distribution; aggregation phenomena.	[36,38]
Co-precipitation	Simple execution; adequate for the synthesis of complex metal oxide NPs; high reproducibility; inexpensive method.	Requires a nanoparticle separation step, for obtaining uniform size distribution; quasi-spherical NPs; risk of oxidation and aggregation phenomena.	[36,39]
Thermal decomposition	Size control; narrow size distribution; crystallinity; Easy scale-up process.	Dilatory process; uses organic solvents; requires further steps to obtain water-soluble MNPs.	[40]
Hydrothermal	Fine particles; no required organic solvents; no required post-treatment; Environmentally benign.	Long reaction times.	[36]
Microemulsification	Simple method; adequate for in vitro and in vivo applications; controllable size and MNP morphology.	Low scalability; reduced quantity of MNPs synthesized; difficult removal of surfactant.	[41]
Polyol-based	Uniform MNPs; size and shape control; simple and reproducible process.	May require high temperature and pressure environment for higher magnetization values.	[42]
Sol-gel	Controlled particle size and shape; production of oxide MNP by gel calcination; adequate for hybrid MNPs.	Requires thermal treatment at elevated temperatures; incomplete removal of matrix components from MNP surface.	[35]
Electrochemical	Ambient temperature environment; narrow size distribution; high purity; adequate for maghemite NPs.	Complicated and long process.	[40,43]
Biosynthesis	High crystallinity; prominent T2 relaxation reduction and contrast.	Reduced control in MNP specifications; mixture of cubic, octahedral and dodecahedral MNPs; low scalability potential.	[37,40]

**Table 3 materials-13-00266-t003:** Magnetic nanoparticles tuned for dual imaging and therapeutic applications.

MNP System Description	Characteristics	Detection Methods	Therapeutic Applications	Tumor	Reference
Gold nanorod-capped magnetite core/mesoporous silica shell nanoparticles	Mean diameter of 386.6 nm; homogenous size distribution; T_2_ relaxivity coefficient of 393.8 mM^−1^·s^−1^; Dox loading capacity of 30% w/w and positive therapy effect under 39–42 °C; no reported cytotoxicity <100 µg/mL; Absorption peak at 790 nm.	MRI	Doxorubicin chemotherapy; PTT	-	[104]
Gold shell-core IONP	Mean diameter: 100 nm; hydrodynamic size: 179 nm; T_2_ relaxivity coefficient of 76.2 mM^−1^·s^−1^	MRI; PAI	PTT	Breast	[105]
Multicore IONP with CuS shell	Mean core diameter of 25.5 nm; hydrodynamic size: 156 nm; zeta potential: −14.1mV at pH 7; magnetization: 84 emu/g;	MRI	MHT; PTT; PDT	-	[31]
cRGD-functionalized Doxorubicin-conjugated and ^64^Cu labelled SPION	Mean core diameter:10 nm; mean hydrodynamic size of the MNP: 68 nm; T_2_ relaxivity coefficient of 101.9 mM^−1^·s^−1^; ^64^Cu T_1/2_:12.7 h; Dox-loading capacity of 5.8% *w*/*w*.	PET; MRI	Doxorubicin chemotherapy	Glioblastoma	[106]
Indium-111 labeled Trastuzumab-Doxorubicin Conjugated, and APTES-PEG coated SPION	Mean diameter: 16 nm; magnetization: 52 emu/g; radiolabel efficiency: 97.6%; trastuzumab conjugation capacity: 63.79%;	SPECT; MRI	Tumor suppression. Antibody and chemotherapeutic agents	Breast	[107]
Manganese-doped iron oxide nanoparticles, coated with bovine serum albumin and functionalized with a cyclic Arg-Gly-Asp (cRGD) peptide and cy5 dye-labelled siRNA	Mean core diameter:15 nm.	MRI	Inhibition of Green fluorescence protein by the siRNA moiety, and interference of receptor-mediated endocytosis via targeting tumor cells overexpressed α_v_β_3_ integrin by RGD peptide.	Breast	[108,109]
Paclitaxel loaded, PEG modified liposome iron oxide MNP	Core size of 7 nm; full nanoplatform size of 168.3 nm; PDI of 0.197; zeta potential of −10.5 mV; paclitaxel entrapment efficiency above 90%.	MRI	Paclitaxel	Breast	[110]
Liposome, ADT loaded iron oxide MNP, encapsulated with PEG	Core size of 7 nm; final size of 211 nm; PDI of 0.19; ADT loading capacity of 49.6%; T_2_* of 12.85 ms;	MRI	H_2_S	Liver	[111]
Rituximab loaded liposome, iron oxide MNP, encapsulated with PEG	Superparamagnetic NP-PVA core size average between 7–10 nm; narrow size distribution (PDI 0.1–0.3); 44.6% SPION-PVA encapsulation efficiency; zeta potential of −9.0 mV.	MRI	Rituximab	Brain Lymphoma	[112]

**Key:** SPION—superparamagnetic iron oxide nanoparticle; MRI—Magnetic resonance imaging; PAI—Photoacoustic imaging; SERS—Surface Enhanced Raman Spectroscopy; APTES—aminopropyl triethoxysiliane; PEG—poly(ethylene glycol); cy5 dye—cyanine dye; cRGD—cyclic arginine-glycine-aspartate peptide; RGD—arginyl-glycyl-aspartic acid; ADT—hydrophobic anethole ditholethione; US—ultrasound; NIR—Near infrared. T_2_*—decay of transverse relaxation, resultant of spin-spin relaxation and inherent inhomogeneity of the main magnetic field.

**Table 4 materials-13-00266-t004:** List, not exhaustive, of clinical trials involving magnetic nanoparticles in cancer diagnosis and/or treatment.

Clinical Trial	Status	MNP	Applications	Tumor	Location
MAGNABLATE I NCT02033447	Completed	IONP for magnetic hyperthermia	Magnetic hyperthermia and MRI	Prostate cancer	University College London Hospital London, UK
NCT01895829	Active	USPIO nanoparticle-ferumoxytol	MRI	Head and neck cancer	University of Texas MD Anderson Cancer Center Houston, TX, USA
NCT00675259	Completed	Paclitaxel albumin-stabilized nanoparticle	Chemotherapy	Breast cancer	Ohio State University Comprehensive Cancer Center Columbus, OH, USA
NCT00920023	Completed	SPIO nanoparticle	MRI	Pancreatic cancer	Massachusetts General Hospital Boston, MA, USA
NCT01927887	Completed	USPIO nanoparticle-ferumoxytol	MRI	Thyroid cancer	Massachusetts General Hospital Boston, MA, USA
NCT01815333	Active	USPIO nanoparticle-ferumoxytol	MRI	Lymph node cancer	University of Texas MD Anderson Cancer Center Houston, TX, USA

Data retrieved from ClinicalTrials.gov [114] Key: SPIO—superparamagnetic iron oxide; USPIO—ultrasmall superparamagnetic iron oxide; MRI—magnetic resonance imaging.

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
