# Peer review of "The Role of Magnetic Nanoparticles in Cancer Nanotheranostics"

_materials, 2020, doi:10.3390/ma13020266_

Round 1

Reviewer 1 Report

Authors nicely reviewed the role of nanoparticles in the field of cancer therapy from fundamental concepts of superparamagnetisem, synthesis methods to functionalization, characterizations and applications of these nanoparticles. Specially, I would appreciate the authors’ efforts to make illustrative tables for comparison.  

Since this is not the only work in this area of science, I would suggest to draw the reader’s attention (probably in the abstract) to the most specific unique idea behind writing this review in comparison with the other review articles of this specific field to make this review more attractive for the readers.

Since XAS and associated XMCD techniques for characterization of magnetic nanoparticles using synchrotron radiation source attracted a lot of attention during the last decades, I would recommend mentioning it in the characterization section.

In order to strengthening the magnetic nanoparticles section, more preliminary magnetism fundamentals is required which could be found in several papers for example check the following one: doi:10.1155/2016/7840161

Although the section of medical applications is nicely described, it could be even more attractive and informative with adding some pictures from the mentioned references.

I strongly recommend expanding the section of hybrid magnetic nanoparticles, as the broad range of multifunctional magnetic nanoparticles of different structures and compositions, which exactly includes the nanotheranostics agents. To this end, authors can consider the following comprehensive review article:doi:10.3390/nano9010097

Author Response

Authors nicely reviewed the role of nanoparticles in the field of cancer therapy from fundamental concepts of superparamagnetisem, synthesis methods to functionalization, characterizations and applications of these nanoparticles. Specially, I would appreciate the authors’ efforts to make illustrative tables for comparison.  

Since this is not the only work in this area of science, I would suggest to draw the reader’s attention (probably in the abstract) to the most specific unique idea behind writing this review in comparison with the other review articles of this specific field to make this review more attractive for the readers.

R: The suggestion of the reviewer was taken into consideration. The abstract now reads:

This review aims at providing a comprehensive description of the current designs, characterization techniques, synthesis methods, and the role of magnetic nanoparticles as promising nanotheranostic agents. A critical appraisal of the impact, potentialities and challenges associated to each technology is also presented.

Since XAS and associated XMCD techniques for characterization of magnetic nanoparticles using synchrotron radiation source attracted a lot of attention during the last decades, I would recommend mentioning it in the characterization section.

R: Information regarding SAS and associated XMCD is now provided in the text. Please see page 9, lines 372-379.

In order to strengthening the magnetic nanoparticles section, more preliminary magnetism fundamentals is required which could be found in several papers for example check the following one: doi:10.1155/2016/7840161

R: Information relative to magnetism fundamentals was added in the text, see page 2, lines 63-79.

Although the section of medical applications is nicely described, it could be even more attractive and informative with adding some pictures from the mentioned references.

R: Following the suggestion of the reviewer Figure 2 was added to the text.

I strongly recommend expanding the section of hybrid magnetic nanoparticles, as the broad range of multifunctional magnetic nanoparticles of different structures and compositions, which exactly includes the nanotheranostics agents. To this end, authors can consider the following comprehensive review article:doi:10.3390/nano9010097

R: The section of hybrid magnetic nanoparticles was duly enlarged. Please see page 7, lines 258-267.

Reviewer 2 Report

This is a timely and important review, especially with the growing interest in nanoparticle technologies for nanomedicine. The order in which the subject material is reviewed is logical and the review itself is comprehensive. I find it especially interesting because the review does not focus on the application of nanoparticle theranostics but does provide a solid resource that covers the synthesis and material properties of such nanoparticles.

I am happy to recommend this review for publication.

Author Response

We appreciate the reviewer's appraisal.

Reviewer 3 Report

Dear authors,

thank you for your interesting review about magnetic nanoparticles for cancer theranostics.

For completion of available methods about particle characterisation (page 6), I would suggest some adds, speaking of Raman spectroscopy for surface molecules and Mössbauer spectroscopy for magnetic core characteristics.

Paragraphs 5. Protection/stabilization and 6. Functionalization; some research groups have worked on thin-coated magnetic nanoparticles (using phosphonate, thiol, carboxylate, ... derived molecules) to overcome limitations of polymer coated nanoparticles (low magnetic core / hydrodynamic radii ratio, sometimes unpredictable colloidal stability, limited diffusion (due to polymer corona thickness), .... I suggest to add some words about this research field.

From a practical point of view, I suggest to facilitate the reading of table 2 (page 4) by horizontal separation of the different methods, by dotted lines or equivalent.

Sincerely yours

Author Response

Dear authors,

thank you for your interesting review about magnetic nanoparticles for cancer theranostics.

For completion of available methods about particle characterisation (page 6), I would suggest some adds, speaking of Raman spectroscopy for surface molecules and Mössbauer spectroscopy for magnetic core characteristics.

R: Information regarding Raman spectroscopy and Mössbauer spectroscopy is now provided, Please see page 9, lines 361-367.

Paragraphs 5. Protection/stabilization and 6. Functionalization; some research groups have worked on thin-coated magnetic nanoparticles (using phosphonate, thiol, carboxylate, ... derived molecules) to overcome limitations of polymer coated nanoparticles (low magnetic core / hydrodynamic radii ratio, sometimes unpredictable colloidal stability, limited diffusion (due to polymer corona thickness), .... I suggest to add some words about this research field.

R: Following the suggestion of the reviewer, Section 5 was enlarged.

From a practical point of view, I suggest to facilitate the reading of table 2 (page 4) by horizontal separation of the different methods, by dotted lines or equivalent.

R: Horizontal separation of the different methods was performed for clarity.

Reviewer 4 Report

This review deals with the usage of magnetic nanoparticles (MNP) for the diagnosis and treatment of different types of tumors. After illustrating the methods to synthesize and characterize MNP, the authors discuss the main strategies to improve MNP bioavailability and efficacy, including functionalization with different polymers and biomolecules. Toxicological aspects and relevant examples of MNP clinical applications are illustrated as well.

The topic is interesting and the paper is well written and well organized, providing the reader with relevant and up-to date information on the use of MNP in the diagnosis and treatment of tumors.

Author Response

We appreciate the reviewer's appraisal.